# MicroRNA-Mediated Downregulation of HMGB2 Contributes to Cellular Senescence in Microvascular Endothelial Cells

**DOI:** 10.3390/cells11030584

**Published:** 2022-02-08

**Authors:** Hye-Ram Jo, Jae-Hoon Jeong

**Affiliations:** 1Division of Radiation Biomedical Research, Korea Institute of Radiological & Medical Science, Seoul 01812, Korea; hrjo6701@gmail.com; 2Radiological and Medico-Oncological Sciences, University of Science and Technology, Daejeon 34113, Korea

**Keywords:** high mobility group box 2, senescence, microRNA, microvascular endothelial cell

## Abstract

High mobility group box 2 (HMGB2) is a non-histone chromosomal protein involved in various biological processes, including cellular senescence. However, its role in cellular senescence has not been evaluated extensively. To determine the regulatory role and mechanism of HMGB2 in cellular senescence, we performed gene expression analysis, senescence staining, and tube formation assays using young and senescent microvascular endothelial cells (MVECs) after small RNA treatment or *HMGB2* overexpression. *HMGB2* expression decreased with age and was regulated at the transcriptional level. siRNA-mediated downregulation inhibited cell proliferation and accelerated cellular senescence. In contrast, ectopic overexpression delayed senescence and maintained relatively higher tube-forming activity. To determine the *HMGB2* downregulation mechanism, we screened miRNAs that were significantly upregulated in senescent MVECs and selected HMGB2-targeting miRNAs. Six miRNAs, miR-23a-3p, 23b-3p, -181a-5p, -181b-5p, -221-3p, and -222-3p, were overexpressed in senescent MVECs. Ectopic introduction of miR-23a-3p, -23b-3p, -181a-5p, -181b-5p, and -221-3p, with the exception of miR-222-3p, led to the downregulation of *HMGB2*, upregulation of senescence-associated markers, and decreased tube formation activity. Inhibition of miR-23a-3p, -181a-5p, -181b-5p, and -221-3p delayed cellular senescence. Restoration of *HMGB2* expression using miRNA inhibitors represents a potential strategy to overcome the detrimental effects of cellular senescence in endothelial cells.

## 1. Introduction

In 1965, Hayflick et al. first used the term “cellular senescence” to describe the limitations of normal human cell proliferation [1]. Cellular senescence is characterized by a stable and irreversible cell cycle arrest, and is associated with multiple cellular and molecular changes. Cellular senescence can compromise tissue repair and regeneration, thereby contributing to aging. Removal of senescent cells can attenuate age-related tissue dysfunction and extend health span [2]. The causes of cellular senescence involve telomere shortening, DNA damage, oxidative stress, and oncogene activation [3]. Telomere shortening generated by repeated DNA replication is mainly responsible for replicative senescence [2,4]. DNA damage induced by radiation and reactive oxygen species resulting in double-stranded DNA break is a potent inducer of stress-induced premature senescence [5,6]. Activation of oncogenes or inactivation of tumor suppressors is a major cause of oncogene-induced senescence [2,6,7]. Activation of the p53/p21^WAF1/CIP1^ and p16^INK4A^/pRB tumor suppressor pathways play a central role in regulating senescence [8].

MicroRNAs (miRNAs) are small, non-coding, endogenous, single-stranded RNA fragments of 22 nucleotides in length. miRNAs regulate gene expression in many cellular processes and associated diseases [9,10,11] by binding to complementary sequences in the 3′-untranslated regions (3′UTRs) of their target mRNAs and inhibiting their gene expression transcriptionally or post-transcriptionally [12]. Recently, miRNAs have been studied as mediators of senescence and aging processes [13]. Many studies have shown that the expression of several miRNAs is altered in senescent human fibroblasts [14,15,16,17], senescent endothelial cells, and other cell types [18,19,20]. miRNAs that act as inducers of senescence are considered as senescence-associated miRNAs (SA-miRNAs) [14,15]. For example, miR-34 decreases SIRT1 expression thereby inducing p53 levels and cell senescence [21,22,23]. miR-195 induces senescence by directly reducing the levels of TERT [24,25]. miR-24 binds the 3′ UTR of p16 mRNA and suppresses its translation in WI-38 HDF and HeLa cells [26].

The vascular endothelial structure consists of endothelial monolayer cells that can exchange substances, such as gases and nutrients, between cells and blood. Endothelial cells represent an important component of this function, and the lack or dysfunction of these cells in the vascular system damages the circulatory system [18]. Cellular senescence in endothelial cells results in endothelial barrier dysfunction and plays an important role in the risk of various cardiovascular diseases [19]. Endothelial function is modulated by traditional cerebrovascular disease risk factors in young adults; however, aging is independently associated with the development of vascular endothelial dysfunction [27,28].

High mobility group box 2 (HMGB2) proteins are the most abundant non-histone chromatin-binding proteins in the nuclei of mammalian cells. It has been proposed that when HMGB proteins bind to chromatin, they locally modify the structure by bending DNA and facilitating the binding of regulatory proteins such as transcription factors, chromatin remodelers, and DNA damage repair machinery [29,30,31]. HMGB proteins control several genomic processes in response to specific biological cues via their interaction with chromatin; therefore, they represent essential regulators of cellular programs as well as disease [32,33,34,35,36]. Previously, we observed that HMGB2 is downregulated by p21 during radiation-induced senescence via the ATM-p53-p21 DNA damage signaling cascade [37]. Therefore, we examined whether HMGB2 is also involved in replicative senescence. In this study, we found that HMGB2 expression decreased with aging, and this regulation was mediated via SA-miRNAs which target HMGB2. Restoration of HMGB2 by targeting these miRNAs represents a potential strategy to overcome the detrimental effects of cellular senescence in endothelial cells.

## 2. Materials and Methods

### 2.1. Cell Culture

Human lung microvascular endothelial cells (MVECs) were purchased from Cell Applications (San Diego, CA, USA) and cultured in EBM-2 media supplemented with EGM-2MV (Lonza, Hopkinton, MA, USA). In total, 5 × 10^5^ cells were plated in a 100 mm culture plate and cultured at 37 °C in a humidified incubator with 5% CO_2_. The cells were passaged every 3 d via trypsinization. Cell counts were determined at the end of every passage after staining with 0.1% trypan blue using a hemocytometer. The cumulative number of population doublings (PDLs) was calculated in relation to the initial cell number.

### 2.2. Telomere Length Assay

Telomere length measurement was performed by quantitative polymerase chain reaction as per the standard procedure 21369534 [38]. Telomere standard curve was generated by plotting threshold cycle (Ct) values against the amount of telomere sequence in kb per reaction suing a ten-fold serial dilution of a synthetic oligonucleotide. Genomic DNA was extracted from MVECs using GeneAll Exgene Cell SV mini kit (GeneAll, Seoul, Korea). Amplification was performed on a Bio-Rad CFX96 machine (Bio-Rad Laboratories, Hercules, CA, USA) in a final volume of 20 µL containing 1× KAPA SYBR^®^ FAST qPCR Master Mix (Kapa Biosystems, Wilmington, MA, USA), 100 nM of both pairs of primers, and 20 ng of genomic DNA. Cycling condition was as follows: 3 min at 95 °C followed by 35 cycles at 98 °C for 7 s and 60 °C for 20 s, followed by a melt curve. Single-copy gene, 36B4, served as a reference gene to normalize the Ct values from the telomere assay. Amplification was performed with 1 μg of genomic DNA as above and cycling condition was as follows: 3 min at 95 °C followed by 35 cycles at 98 °C for 7 s and 58 °C for 20 s. Absolute telomere length was calculated and the results are expressed in kb/genome.

### 2.3. Western Blot Analysis

Cells were lysed in RIPA buffer (20 mM Tris-HCl [pH 7.5], 150 mM NaCl, 1 mM EDTA, 1 mM EGTA, 1% NP-40, 1% sodium deoxycholate, and protease inhibitors) and briefly sonicated. Protein content was measured using the Coomassie (Bradford) protein assay kit (Thermo Fisher Scientific, Waltham, MA, USA), and equal amounts of cell lysate were separated via SDS-polyacrylamide gel electrophoresis and transferred to nitrocellulose membranes (GE Healthcare Biosciences, Foster City, CA, USA). Membranes were immunoblotted with antibodies against HMGB2 (Abcam, Cambridge, UK), p16, p21, and β-actin (Santa Cruz Biotechnology, Dallas, TX, USA), and detected via chemiluminescence using ECL detection reagents.

### 2.4. Quantitative Real-Time Polymerase Chain Reaction (qRT-PCR)

For the analysis of mRNA expression, total RNA was isolated using a Hybrid-R Total RNA Purification kit (GeneAll). cDNA was synthesized using the PrimeScript RT Master Mix kit (TaKaRa Bio, Kusatsu, Japan), and quantitative PCR was performed using the qPCR Green 2X master mix (MBiotech, Seoul, Korea) on a Bio-Rad CFX96 machine (Bio-Rad Laboratories). Gene expression was normalized to that of two reference genes (PPIA and RPL13A), and the relative gene expression values were calculated based on the Ct value using the 2-ΔΔCt method [32]. For the analysis of miRNA expression, total RNA was isolated using the TRIzol reagent (Thermo Fisher Scientific) and cDNA was synthesized using the Mir- X™ miRNA First-Strand Synthesis Kit (Clontech Laboratories, Palo Alto, CA, USA). Quantitative PCR was performed using the qPCR Probe 2× Master Mix (MBiotech, Seoul, Korea). The primers used for qRT-PCR are listed in Appendix A.

### 2.5. Senescence-Associated β-Galactosidase (SA-β-gal) Staining

Cellular SA-β-gal activity was measured as described previously [33]. Briefly, 1.5 × 10^5^ cells were plated in a 60 mm dish and cultured for 3 d. Cells were washed with phosphate-buffered saline (PBS) and fixed with 2% (*v/v*) paraformaldehyde and 2% (*v/v*) glutaraldehyde in PBS for 10 min at room temperature. The presence of SA-β-gal activity was determined by incubating the cells in a solution containing 40 mM citric acid-sodium phosphate (pH 6.0), 150 mM NaCl, 2 mM MgCl_2_, 5 mM potassium ferricyanide, 5 mM potassium ferrocyanide, and 1 mg/ml X-gal for 14 h at 37 °C in a dark incubator. Cells were counterstained with eosin solution (100 mg eosin Y, 0.5% (*v/v*) acetic acid in 80% ethyl alcohol) and the proportion of blue cells observed under a light microscope was measured.

### 2.6. Tube Formation Assay

Tube formation was evaluated as described previously [34,35]. Briefly, 5 × 10^4^ MVECs were seeded onto each well of 24-well plates pre-coated with BD Matrigel Matrix (BD Biosciences, San Jose, NJ, USA) and incubated overnight to allow the formation of tube-like structures. Endothelial cell tube formation was assessed using a GE InCell Analyzer 2000 (GE Healthcare Life Sciences, Little Chalfont, UK) and quantified using Image J software (National Institutes of Health, Bethesda, MD, USA) using the Angiogenesis Analyzer plugin.

### 2.7. Transfection of Small Interfering RNA (siRNA), miRNA, and miRNA Inhibitors

The siRNAs and synthetic miRNA mimics used in this study were synthesized by Genolution Pharmaceuticals (Seoul, Korea). miRNA inhibitors involving 2′-O-methyl-modified oligoribonucleotide single strands were purchased from Genolution Pharmaceuticals and Integrated DNA Technologies (Singapore). Briefly, 1.5 × 10^5^ cells were reverse-transfected with 20 nM siRNA/miRNA or 40 nM miRNA inhibitor using Lipofectamine RNAiMAX transfection reagent (Thermo Fisher Scientific, Waltham, MA, USA) in a 60 mm dish, according to the manufacturer’s instructions. The sequences of the siRNA and miRNA mimics are listed in Appendix A.

### 2.8. Overexpression of HMGB2

Ectopic overexpression of *HMGB2* was performed as described previously [36]. MVECs were transduced with pLNCX or pLNCX-HMGB2. After 2 d of incubation, transduced cells were aliquoted for western blotting, SA-β-gal staining, proliferation assay, and tube formation assay.

### 2.9. Small RNA Sequencing

Total RNA was extracted from young (PDL23) and old (PDL49) MVECs using TRIzol reagent (Thermo Fisher Scientific, Waltham, MA, USA). RNA quality check, library preparation, next generation sequencing, and data analysis were performed by ebiogen (Seoul, Korea), and the small RNA sequencing data were deposited in the GEO database (GSE192677).

### 2.10. Luciferase Reporter Assay

The 3′UTR region of the human *HMGB2* gene (NM_001130688.1) was subcloned into the 3′ region of the luciferase gene of the pGL3UC luciferase reporter vector [39]. The primer sequences for PCR amplification were 5′-AAACTCTCTAGAATGGCTATCCTTTAATGATGC-3′ and 5′-AATGCCCTGCAGACACCTGAGGAACAATTTAG-3′. The 293T cells were seeded in 24-well plates and co-transfected with reporter plasmid pGL3UC-HMGB2 3′UTR (200 ng), pRL-CMV-Renilla plasmid (2 ng), and miRNAs (20 nM) using Lipofectamine™ 2000 (Thermo Fisher Scientific, Waltham, MA, USA). After 48 h, luciferase activity was measured using the Dual-Luciferase Reporter Assay system (Promega, Madison, WI, USA) according to the manufacturer’s instructions. Firefly luciferase activity was normalized to Renilla luciferase activity.

### 2.11. Statistical Analysis

Data are presented as the mean ± standard error. All results were analyzed for statistical significance using one-way and two-way analysis of variance (ANOVA) with Fisher’s PLSD post-hoc test. A *p*-value of 0.01 (**) and 0.001 (***) was compared with that of the control group. Statistical analyses were performed using StatView version 5.0.1.

## 3. Results

### 3.1. HMGB2 Expression Decreased in Aged MVECs

To determine the role of HMGB2 in the aging of microvessels and aging-related vascular disease, we used primary MVEC cultures as a replicative senescence model. Replicative senescence was induced by continuously subculturing early passaged MVECs at PDL20 until they reached the end of their lifespan (PDL51) (Figure 1A). The absolute telomere length (aTL) was also measured to demonstrate that senescence was indeed triggered by telomere shortening. As cell proliferation was found to slow down from PDL42 onwards, we considered that cellular senescence was initiated from this point for convenience. Young (PDL28), early senescent (PDL45), and fully senescent (PDL51) MVECs were subjected to SA-β-Gal histochemical staining (Figure 1B). Senescent MVECs exhibited poor growth and senescent morphology with strong SA-β-Gal staining. Under the same conditions, we measured HMGB2 protein levels using western blot analysis (Figure 1C). Similar to other types of cells, HMGB2 levels were decreased and the expression of two canonical senescence markers, p16 and p21, was induced during cellular senescence. To verify whether this decrease in HMGB2 level during cellular senescence is regulated at the transcriptional level, qRT-PCR was performed. *HMGB2* mRNA levels gradually decreased with the population doubling, showing an inverse correlation with the transcription of p16 and p21 genes (Figure 1D).

### 3.2. HMGB2 Silencing Induced Premature Senescence of MVECs

To determine the effect of *HMGB2* downregulation on cellular senescence, *HMGB2* was downregulated in young MVECs (PDL28) via treatment with siRNA which targeted *HMGB2*. After transfection, *HMGB2* knockdown was confirmed via western blot analysis (Figure 2A). *HMGB2* knockdown was associated with classical features of cellular senescence in young MVECs, as demonstrated by an impaired cell proliferative potential (Figure 2B) and an increased proportion of SA-β-gal-positive cells (Figure 2C). Tube formation assay is one of the most widely used in vitro assays to model the reorganization of angiogenesis, and measures the ability of endothelial cells to form capillary-like structures. *HMGB2* knockdown caused MVECs to form a tube structure in a less efficient manner than that observed in the control (Figure 2D). These results suggest that HMGB2 depletion contributes to premature senescence in MVECs.

### 3.3. HMGB2 Overexpression Delayed Replicative Senescence in MVECs

Next, we investigated whether ectopic overexpression of *HMGB2* can delay senescence progression. Retrovirus-mediated *HMGB2* overexpression was established in exponentially growing MVECs (PDL30) and serially passaged cells. *HMGB2* overexpression was maintained in serially passaged cells (PDLs 36–45) (Figure 3A). Overexpression of *HMGB2* did not substantially increase the proliferative potential of cells compared to that in the vector-only control (Figure 3B), although it could not rejuvenate MVECs (Figure 2B). In addition, *HMGB2* overexpression decreased the proportion of SA-β-gal-positive cells (Figure 3C) and maintained relatively better tube formation activity (Figure 3D). Therefore, ectopic overexpression of HMGB2 may help MVECs maintain a healthy state.

### 3.4. Screening of SA-miRNAs That Target HMGB2 in MVECs

Growing evidence suggests that miRNAs act as inducers of senescence [40,41]. Therefore, we speculated that miRNAs targeting *HMGB2* may be induced which downregulate *HMGB2*, leading to senescence in MVECs. As senescent cells of different lineages demonstrate tissue-specificity in miRNA profiles [42], we performed small RNA sequencing to screen for miRNAs specifically induced in senescent MVECs (PDL49) relative to proliferating cells (PDL23). miRNA candidates that targeted HMGB2 were searched based on the 3′UTR sequence of *HMGB2* using seven sequence-based miRNA search platforms (Appendix A). All miRNA candidates were pooled (*n* = 75), and we selected miRNAs recommended by more than three platforms (*n* = 25) (Appendix A). Among them, six miRNAs with sufficient read numbers were selected for further analysis (Figure 4A). The sequence alignment of miRNAs with the possible target sequences in the 3′UTR of *HMGB2* are shown in Figure 4B. The expression of selected miRNAs in serially passaged MVECs was analyzed via qRT-PCR, which confirmed that all six miRNAs were induced in senescent MVECs (Figure 4C).

### 3.5. HMGB2-Tageting SA-miRNAs Regulated Cellular Senescence in MVECs

To identify miRNAs that downregulate *HMGB2* in vivo, we transfected the synthetic miRNA mimics into MVECs and performed western blotting (Figure 5A). Among the six candidates, all candidates except miR-222-3p downregulated *HMGB2*, which confirmed that the *HMGB2* gene is the target of these miRNAs in MVECs. Next, we confirmed the target specificity of miRNAs using the luciferase reporter system since miR-221-3p and miR-222-3p share a common target sequence (Figure 4B). All miRNAs except for miR-222-3p repressed luciferase activity (Figure 5B) when the luciferase reporter system containing the 3′UTR sequence of *HMGB2* was co-transfected with miRNA. We then confirmed that the introduction of exogenous miRNA accelerated cellular senescence, and the proportion of SA-β-gal-positive cells increased (Figure 5C). In addition, five miRNA candidates inhibited angiogenic tube formation (Figure 5D). These results suggest that miR-23a-3p, miR-23b-3p, miR-181a-5p, miR-181b-5p, and miR-221-3p represent SA-miRNAs that target HMGB2 in MVECs.

### 3.6. Inhibition of HMGB2-Targetting SA-miRNAs Delayed Senescence in MVECs

To confirm the effects of the selected miRNAs on cellular senescence, we used a miRNA inhibitor, a steric-blocking oligonucleotide that hybridizes to mature miRNAs and inhibits their function. Transfection of miRNA inhibitors into MVECs (PDL42) was associated with relatively higher expression levels of HMGB2 protein in senescent MVECs (Figure 6A), which may be due to the prevention of *HMGB2* downregulation induced by miRNA. As expected, these MVECs displayed a lower level of cellular senescence, a lower proportion of SA-β-gal-positive cells (Figure 6B), and higher angiogenic tube formation activity (Figure 6C) than those observed in cells treated with miRNA alone. These results suggest that miRNA-mediated downregulation of *HMGB2* contributes to cellular senescence, and targeting miRNAs can delay senescence progression.

## 4. Discussion

HMGB2 is the most abundant non-histone chromatin-binding protein present in the nuclei of mammalian cells [43]. HMGB proteins are known to modulate the local chromatin environment, facilitating the binding of other proteins to chromatin, and controlling nuclear processes such as transcription, DNA damage repair, and nucleosome sliding [44]. Growing evidence supports the key role of HMGB2 in cellular senescence and related diseases. Aging-related loss of HMGB2 in articular cartilage is linked to reduced cellularity, which contributes to the development of osteoarthritis [45]. HMGB2 is involved in cell cycle arrest and chromatin remodeling during senescence. For example, HMGB2 binds to the senescence-associated secretory phenotype gene loci and protects loci from gene silencing via heterochromatin spreading [46]. A recent study has shown that HMGB2 is depleted from the nucleus upon initiation of cellular senescence, resulting in the reorganization of the genome and changes in transcriptional activity. HMGB2 modulates the global chromatin structure and expression of genes found within topologically associating domains by insulating against the clustering of CTCF proteins [47]. These studies support the crucial role of HMGB2 in senescence and aging processes. However, the mechanism of *HMGB2* downregulation upon initiation of senescence has not yet been addressed. Studying the downregulation mechanism is important for controlling senescence and preventing the detrimental effects of senescence. In the present study, we showed that age-associated induction of miRNAs targeting HMGB2 contributes to the depletion of HMGB2 during replicative senescence in MVECs.

The overall HMGB2 level decreased with age (Figure 1C), mostly at the transcriptional level (Figure 1D). Although p21 inhibits *HMGB2* transcription during radiation-induced senescence [37], p21 does not seem to be crucial in replicative senescence since HMGB2 depletion is followed by induction of p21. This signifies that the canonical DNA damage signaling, mediated by the ATM-p53 signaling pathway, does not significantly contribute to the downregulation of *HMGB2*, even if telomere erosion and subsequent DNA damage are important triggering events in replicative senescence. However, two canonical CDK inhibitors, p16 and p21, may play a role in accelerating the depletion of HMGB2 via cell cycle regulation. As E2F1 controls the timely expression of the S phase of the cell cycle-specific genes, including *HMGB2* [48], activation of these CDK inhibitors and cell cycle arrest at the G1 phase inevitably leads to a decrease in *HMGB2* transcription. Therefore, we speculate that the inhibition of *HMGB2* transcription could have been an accompanying result of cell cycle arrest, and other mechanisms downregulating *HMGB2* may function upon initiation of senescence.

In this study, we identified five SA-miRNAs targeting HMGB2 in MVECs: miR-23a-3p, miR-23b-3p, miR-181a-5p, miR-181b-5p, and miR-221-3p (Figure 5). Although these miRNAs have been reported to regulate senescence in other types of cells or angiogenesis in HUVEC, HMGB2 has not been considered a prime target of these miRNAs. While miR-23a-3p is reported to regulate dermal aging and senescence by targeting hyaluronan synthase 2 (*HAS2*) [49], miR-181a/b-5p contributes to the induction of senescence in primary human keratinocytes by targeting sirtuin 1 (*SIRT1*) [50]. Markopoulos et al. presented a distinct set of 15 miRNAs, including these five miRNAs, that are significantly upregulated in senescent human lung fibroblasts [51]. Based on pathway analysis of miRNA target genes, this subset of miRNAs acts in concert to induce cell cycle phase arrest and telomere erosion, establishing a senescent phenotype. In addition, miR-181a/b and miR-221 have been reported to modulate angiogenesis by targeting platelet-derived growth factor receptor A (*PDGFRA*) [52], and tissue inhibitor of metalloproteinase 3 (*TIMP3*) and hypoxia-inducible factor 1-α (*HIF1A*) [53,54], respectively. miR-222-3p inhibits migration and tube formation in endothelial progenitor cells by targeting adiponectin receptor 1 (*ADIPOR1*) [55]. Fibroblasts and endothelial cells have a common subset of miRNAs that can regulate senescence, even though specific and distinct expression of miRNAs has been reported.

Recently, HMGB2-targeting miRNAs have been reported to regulate various biological functions. Since it is important to maintain a sufficient intranuclear amount of HMGB2 protein for the proper function of HMGB2, the downregulation via miRNA can induce related disorders. miRNA-127 controls embryonic stem cell pluripotency via the HMGB2-OCT/SOX2 axis [56], and miR-590-3p contributes to the severity of IgA nephropathy via downregulation of *HMGB2* in peripheral mononuclear cells [57]. The abnormal overexpression of miR-23b-5p promotes cardiac hypertrophy and dysfunction via HMGB2 [58], and aberrant expression of miR-127-5p impairs the function of granulosa cells via *HMGB2* in premature ovarian insufficiency [59]. In glioma cells, miR-130a negatively regulates the oncogenic functions of HMGB2 [60], and miR139-5p regulates the progression of osteosarcoma by modulating *HMGB2* expression [61]. miR-23b-3p regulates the chemoresistance of gastric cancer cells by targeting autophagy-related gene 12 (*ATG12*) and HMGB2 [62]. miRNA-130a-5p suppresses myocardial ischemia reperfusion injury by downregulating the HMGB2/NF-κB axis [63].

Senescent cells accumulate in tissues and cause age-related decline and disease with aging. Studies show that there is a relationship between HMGB2 and cardiovascular diseases in cardiac myocytes [64,65]. Senescence in tissues is affected by the interaction with neighboring cells in the local microenvironment [66]. The findings of the present study can help prevent cardiovascular diseases and other related diseases. Restoring HMGB2 activity by using miRNA inhibitors can overcome the detrimental effects of aging and pathological remodeling.

## Figures and Tables

**Figure 1 cells-11-00584-f001:**
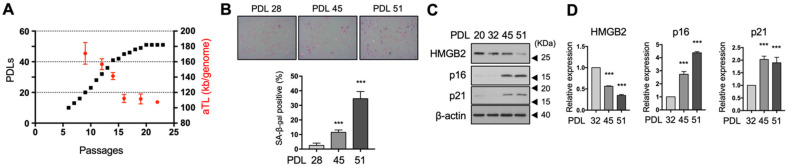
*HMGB2* expression was decreased in replicative senescent MVECs. (**A**) The cumulative number of PDLs was calculated by counting the viable cells during subculturing and a PDL curve was generated. Telomere length per genome were also measured. (**B**) Senescence in MVECs was evaluated via SA-β-gal staining and the proportion of SA-β-gal-positive cells is shown. (**C**) HMGB2 protein levels were determined via immunoblotting analysis using MVECs showing different PDLs. The expression levels of canonical senescence markers, p16 and p21, are shown simultaneously. (**D**) The mRNA expression levels of *HMGB2*, p16, and p21 were quantified via quantitative real time-polymerase chain reaction. HMGB2, high mobility group box 2; MVECs, microvascular endothelial cells; PDL, population doubling; aTL, absolute telomere length; SA-β-gal, senescence-associated beta galactosidase. *** *p* < 0.001.

**Figure 2 cells-11-00584-f002:**
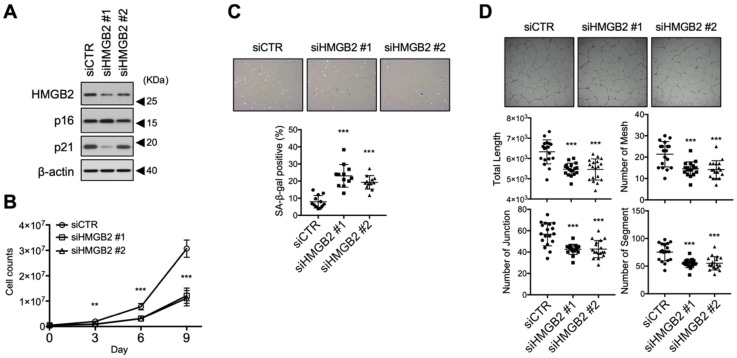
*HMGB2* silencing induced premature senescence in MVECs. (**A**) Proliferating MVECs (PDL23) were transfected with two siRNAs designed to target HMGB2. Knockdown of *HMGB2* was confirmed via immunoblot analysis. The expression levels of p16 and p21 are shown simultaneously. (**B**) Cell proliferation was determined by counting the viable cells after trypan blue staining every 3 d. (**C**) Effect of *HMGB2* silencing on cellular senescence was analyzed via SA-β-gal staining and the proportion of SA-β-gal-positive cells is shown. (**D**) The effect of *HMGB2* knockdown on the angiogenic activity of MVECs was analyzed via in vitro tube formation assay and quantified using Image J software with the Angiogenesis Analyzer plugin. siCTR, small interfering RNA control; siHMGB2, small interfering RNA against *HMGB2.* ** *p* < 0.01, *** *p* < 0.001.

**Figure 3 cells-11-00584-f003:**
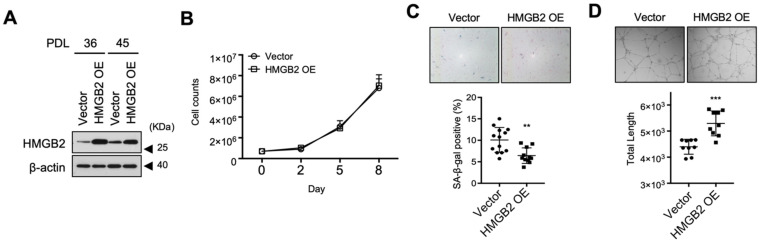
*HMGB2* overexpression delayed replicative senescence in MVECs. (**A**) *HMGB2* was overexpressed in MVECs (PDL30) using a retrovirus system and passaged serially to PDL45. Overexpression of *HMGB2* was confirmed via immunoblot analysis. (**B**) Cell proliferation was determined by counting the viable cells after trypan blue staining every 3 d. (**C**) Senescence in MVECs (PDL41) was determined via SA-β-gal staining and the proportion of SA-β-gal-positive cells was compared statistically. (**D**) The effect of *HMGB2* overexpression on angiogenic potential was analyzed via in vitro tube formation assays. Representative images are shown and the total length of tube-like structures in a field was statistically compared. HMGB2 OE, *HMGB2* overexpression. ** *p* < 0.01, *** *p* < 0.001.

**Figure 4 cells-11-00584-f004:**
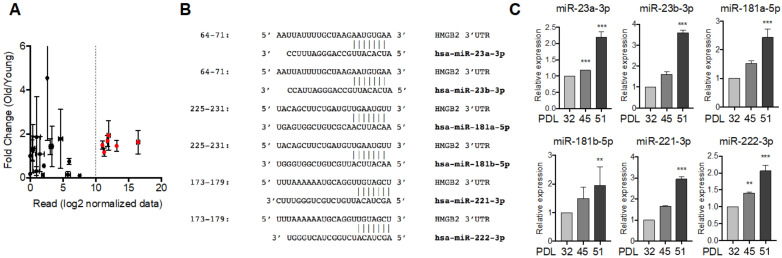
Screening of miRNAs targeting HMGB2 and induced during senescence in MVECs. (**A**) Twenty-five candidate miRNAs that targeted *HMGB2* were plotted using the normalized expression level as the x-axis and the induction fold of expression in old MVECs as the y-axis. Red dots represent the finally selected miRNAs. (**B**) Sequence alignment of miRNA with the possible target sequence in the 3′UTR of *HMGB2*. Numbers indicated the location from the stop codon of the *HMGB2* gene. (**C**) Expression of miRNAs was analyzed by qRT-PCR in serially passaged MVECs. 3′UTR, 3′ untranslated region. ** *p* < 0.01. *** *p* < 0.001.

**Figure 5 cells-11-00584-f005:**
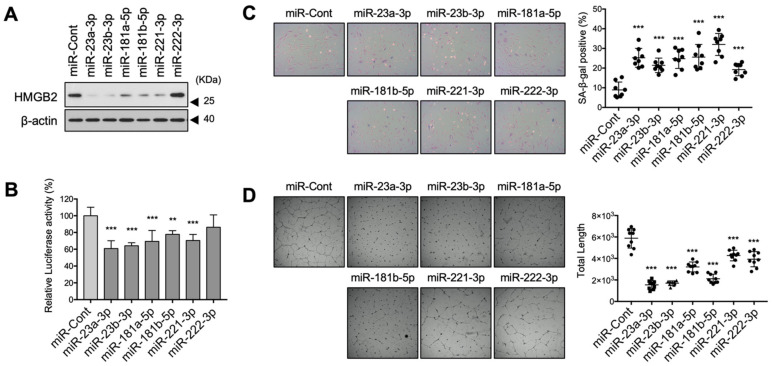
MicroRNAs targeting HMGB2 regulated cellular senescence in MVECs. (**A**) Proliferating MVECs (PDL28) were transfected with the selected miRNA mimics, and the HMGB2 protein level was examined via immunoblot analysis. (**B**) Luciferase activities were measured after co-transfection with *Luc* reporter gene fused with 3′UTR sequence of HMGB2 and the indicated miRNAs into 293T cells. (**C**) Senescence in MVECs was determined via SA-β-gal staining and the proportion of SA-β-gal-positive cells was compared statistically. (**D**) The effect of miRNA mimics on the angiogenic activity of MVECs was analyzed via in vitro tube formation assays and the total length of tube-like structures in a field was statistically compared. ** *p* < 0.01, *** *p* < 0.001.

**Figure 6 cells-11-00584-f006:**
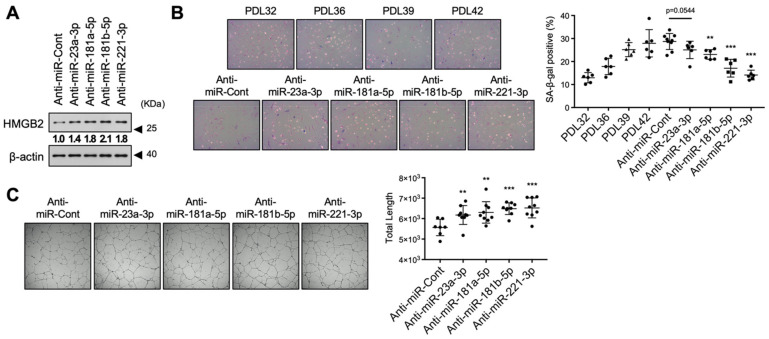
Treatment with miRNA inhibitors delayed cellular senescence in MVECs. (**A**) MVECs (PDL42) were transfected with miRNA inhibitors, and HMGB2 protein levels were determined via immunoblot analysis. Numbers in bold indicates the relative amount of HMGB2 normalized with β-actin. (**B**) Senescence in MVECs was determined via SA-β-gal staining and the proportion of SA-β-gal-positive cells was compared statistically. (**C**) The effect of miRNA mimics on the angiogenic activity of MVECs was analyzed via in vitro tube formation assays, and the total length of tube-like structures in a field was statistically compared. ** *p* < 0.01. *** *p* < 0.001.

## Data Availability

The data presented in this study are available upon request from the corresponding author.

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
