# Peer review of "MicroRNA-Mediated Downregulation of HMGB2 Contributes to Cellular Senescence in Microvascular Endothelial Cells"

_cells, 2022, doi:10.3390/cells11030584_

Round 1

Reviewer 1 Report

In this study, the authors showed that HMGB2 expression is decreased in senescent MVECs and that overexpression of HMGB2 seems to delay the onset of senescence. They identified 5 SA-miRNAs which targets HMGB2.

Introduction:

1/ The English needs to be improved.

2/ Quote more recent and pertinent papers, specially for the part regarding the molecular pathways triggering cellular senescence. The part regarding the signalling pathways involved in the induction of replicative senescence vs stress induced senescence is oversimplified and not completely exact.

3/ The role of SA-miRNAs should be more detailed. Give examples of targets of these miRNAs and how they induce senescence.

Results:

1/ The authors should assess the telomere length of MVECs at few time points of the growth curve (e.g. at the beginning of the cell culture and before senescence) to demonstrate that senescence is indeed triggered by telomere shortening and not by the conditions of culture.

2/ line 183 and line 184: replace “aging” by “cellular senescence”. And line 185, replace “with age” by “with the population doublings”. Cellular senescence and aging are not the same.

3/ In SiHMGB2 MVECs check also the levels of p16 and p21 by WB.

4/ Figure 3B: Show the growth curves (cumulative PDs as a function of days) instead of cell counts. Line 221: the cell counts don’t show a “substantial” increase in proliferation after HMGB2 overexpression. The proliferation assay should be done at least twice to be able to determine the statistical significance.

Author Response

Reviewer 1

In this study, the authors showed that HMGB2 expression is decreased in senescent MVECs and that overexpression of HMGB2 seems to delay the onset of senescence. They identified 5 SA-miRNAs which targets HMGB2.

Introduction:

1/ The English needs to be improved.

>> Manuscript was reviewed by a professional editing service and revised accordingly (not specified).

2/ Quote more recent and pertinent papers, specially for the part regarding the molecular pathways triggering cellular senescence. The part regarding the signalling pathways involved in the induction of replicative senescence vs stress induced senescence is oversimplified and not completely exact.

>> The first part of introduction regarding cellular senescence was revised and updated as recommended (Page 1, Lines 29-39).

3/ The role of SA-miRNAs should be more detailed. Give examples of targets of these miRNAs and how they induce senescence.

>> The role of miRNAs in senescence induction was added in detail as recommended. (Page 2, Lines 48-52).

Results:

1/ The authors should assess the telomere length of MVECs at few time points of the growth curve (e.g. at the beginning of the cell culture and before senescence) to demonstrate that senescence is indeed triggered by telomere shortening and not by the conditions of culture.

>> The absolute telomere length (aTL) was measured to demonstrate that senescence was indeed triggered by telomere shortening. Figure 1(B) was modified with aTL as a second Y-axis and text was rewritten accordingly (Pages 2-3, Lines 85-99; Page 4, Lines 190-192; Page 5, Line 204, Figure 1 (A), Lines207, 212-214). Standard curve of telomere and SCG used to calculate aTL are shown below.

2/ line 183 and line 184: replace “aging” by “cellular senescence”. And line 185, replace “with age” by “with the population doublings”. Cellular senescence and aging are not the same.

>> Manuscript was rewritten for proper description as recommended. (Pages 4-5, Lines 199, 200, and 202).

3/ In SiHMGB2 MVECs check also the levels of p16 and p21 by WB.

>> The protein level of p16 and p21 were also explored in siRNA-treated samples and Figure 2(A) was updated as recommended (Page 5, Line 210, Figure 2A).

4/ Figure 3B: Show the growth curves (cumulative PDs as a function of days) instead of cell counts. Line 221: the cell counts don’t show a “substantial” increase in proliferation after HMGB2 overexpression. The proliferation assay should be done at least twice to be able to determine the statistical significance.

>> There was a typo error in Line 242. Manuscript was rewritten as follow. “Overexpression of HMGB2 did not substantially increased the proliferative potential of cells ~” (Page 6; Line 226). In addition, proliferation assay was repeated and analyzed statistically. Error bars were added in Figure 3B (Page 6, line 231).

Although we redraw the graph with PDLs on the Y-axis, the cell counts version is more informative.

Reviewer 2 Report

In this work, authors analysed the role of HMGB2 in replicative senescence in microvascular endothelial cells also suggesting a modulation of HMGB2 expression by specific microRNAs. 

I have some comments

  • the luciferase assay should demonstrate a direct regulation of candidate microRNAs and HMGB2 but there are no significant differences among miRNAs and control in the luciferase activity (fig 5B). Did the authors forget to insert asterisks? This point is the most controversial since the analysed miRNAs have already a role in senescence as well as HMGB2 and the novelty of paper is lost without the demonstration of a strong direct correlation. 
  • As regard SA-miRNA inhibition I think the comparison with early senescent and fully senescent cells would be more informative than young one (Fig 6B). Indeed, PDL45 cells show an early senescent phenotype and the phenotype of PDL42 treated with inhibitors should be compared with them to support the idea of senescence prevention. In addition, did authors consider to evaluate beta Gal positive cells and angiogenic activity at different time point of treatment? 
  • The figure 6A should be implemented with a graphic and relative statistics to evaluate the real differences among anti-miR-Cont and the candidate miRNAs
  • Why authors did not evaluate SASP expression since they suggest a HMGB2-driven mechanism in secretory phenotype modulation?

Author Response

Reviewer 2

In this work, authors analysed the role of HMGB2 in replicative senescence in microvascular endothelial cells also suggesting a modulation of HMGB2 expression by specific microRNAs.

I have some comments

  • the luciferase assay should demonstrate a direct regulation of candidate microRNAs and HMGB2 but there are no significant differences among miRNAs and control in the luciferase activity (fig 5B). Did the authors forget to insert asterisks? This point is the most controversial since the analysed miRNAs have already a role in senescence as well as HMGB2 and the novelty of paper is lost without the demonstration of a strong direct correlation.

>> The luciferase assay was repeated and statistically analyzed, and the Figure 5B was updated accordingly (Page 8, line 306, Figure 5B). Only miR-222-3p did not downregulate HMGB-Luc significantly, which coincides with the western blot result (Figure 5A). Both results support the importance of HMGB2 and the specificity of miRNAs in regulation of senescence.

  • As regard SA-miRNA inhibition I think the comparison with early senescent and fully senescent cells would be more informative than young one (Fig 6B). Indeed, PDL45 cells show an early senescent phenotype and the phenotype of PDL42 treated with inhibitors should be compared with them to support the idea of senescence prevention. In addition, did authors consider to evaluate beta Gal positive cells and angiogenic activity at different time point of treatment?

>> We included the SA-beta-Gal assay result of MVECs of different PDLs as controls instead of just the young one in Figure 6B as recommended (Page 9, line 315, Figure 6B). We optimized the time point for SA-beta-Gal assay and tube formation assay, which included the results.

  • The figure 6A should be implemented with a graphic and relative statistics to evaluate the real differences among anti-miR-Cont and the candidate miRNAs

>> Intensity of protein bands was measured via image analysis software, and the relative amount of HMGB2 normalized to beta-actin was presented in Figure 6A (Page 9, Line 315, Figure 6A, Lines, 318-319).

  • Why authors did not evaluate SASP expression since they suggest a HMGB2-driven mechanism in secretory phenotype modulation?

>> In my opinion, HMGB2 depletion is one thing and SASP gene regulation by HMGB2 is another. Although these two HMGB2-related phenomena occurred during the early state of senescence, there is no evidence that these two events are interrelated or associated biologically. I think that HMGB2 depletion may be the result of cell cycle arrest, transcription inactivation, and downregulation by miRNAs. Preventing heterochromatin spreading could be one of the roles of HMGB2 in the onset of senescence, as has been reported by several recent and excellent papers (46, 47). So, in this paper, I focused on the downregulation mechanism of HMGB2, especially by SA-miRNAs, and its biological significance in senescence.

As another project, we examined SASP gene expression in senescent and siHMGB2 treated MVECs. However, as shown below, not all SASP genes were regulated in the same way and the contribution of HMGB2 on SASP gene expression seemed different. We thought that evaluating SASP gene expression as a marker of senescence could make the story too complicated, so we did not include SASP gene regulation in this study.

Round 2

Reviewer 1 Report

The revised version addressed all the points highlighted in the first round of revisions.

Reviewer 2 Report

Thanks for your answers and improvements.